# Advances in Multimodal Emotion Recognition Based on Brain–Computer Interfaces

**DOI:** 10.3390/brainsci10100687

**Published:** 2020-09-29

**Authors:** Zhipeng He, Zina Li, Fuzhou Yang, Lei Wang, Jingcong Li, Chengju Zhou, Jiahui Pan

**Affiliations:** 1School of Software, South China Normal University, Foshan 528225, China; zhipenghe@m.scnu.edu.cn (Z.H.); fuzhouyang@m.scnu.edu.cn (F.Y.); wezard@m.scnu.edu.cn (L.W.); lijingcong@hotmail.com (J.L.); cjZhou@scnu.edu.cn (C.Z.); 2School of Computer, South China Normal University, Guangzhou 510641, China; zina@m.scnu.edu.cn

**Keywords:** emotion recognition, multimodal fusion, brain–computer interface (BCI), affective computing

## Abstract

With the continuous development of portable noninvasive human sensor technologies such as brain–computer interfaces (BCI), multimodal emotion recognition has attracted increasing attention in the area of affective computing. This paper primarily discusses the progress of research into multimodal emotion recognition based on BCI and reviews three types of multimodal affective BCI (aBCI): aBCI based on a combination of behavior and brain signals, aBCI based on various hybrid neurophysiology modalities and aBCI based on heterogeneous sensory stimuli. For each type of aBCI, we further review several representative multimodal aBCI systems, including their design principles, paradigms, algorithms, experimental results and corresponding advantages. Finally, we identify several important issues and research directions for multimodal emotion recognition based on BCI.

## 1. Introduction

Emotion is a general term for a series of subjective cognitive experiences. Emotions consist of a set of psychological states generated by various feelings, thoughts and behaviors. People convey emotional information constantly during the process of communication; emotion recognition plays an important role in interpersonal communication and many aspects of daily life. For example, recognizing the emotional states of patients with emotional expression disorders would be helpful in supplying better treatments and care. Emotion recognition is also an indispensable aspect of humanization in human-computer interaction. The advent and development of portable noninvasive sensor technologies such as brain–computer interfaces (BCI) provide effective methods for achieving humanization in human-computer interactions.

A BCI consists of technology that converts signals generated by brain activity into control signals for external devices without the participation of peripheral nerves and muscles [1]. Affective BCI (aBCI) [2] originated from a research project in the general communication field that attempted to create neurophysiological devices to detect emotional state signals and then to use the detected information to promote human-computer interaction. aBCI uses techniques from psychological theories and methods (concepts and protocols), neuroscience (brain function and signal processing) and computer science (machine learning and human-computer interaction) to induce, measure and detect emotional states and apply the resulting information to improve interaction with machines [3]. Research in the aBCI field focuses on perceiving emotional states, modeling of emotional processes, synthesizing emotional expression and behavior and improving interactions between humans and machines based on emotional background [4]. In this study, we focus primarily on emotion recognition in aBCI systems.

In emotion-recognition research, one of the most important problems is to describe the emotional state scientifically and construct an emotional description model. Emotional modeling involves establishing a mathematical model to describe an emotional state, which can then be classified or quantified by an aBCI system. Establishing an emotion model is an important aspect of emotional measurement because it allows us to make more accurate assessments of emotional states. Many researchers have proposed emotional representation methods; these can be divided into discrete emotional models and dimensional emotional models.

For discrete emotion model, emotional states are composed of several basic discrete emotions, (e.g., the traditional concept of “joys and sorrows”). Ekman [5] suggested that emotions include sadness, fear, disgust, surprise, joy and anger and that these six basic emotions can form more complex emotional categories via combined patterns. However, this description neither describes the connotation of emotion scientifically nor enables a computer to analyze emotional state from a computational point of view. The dimensional emotional model maps emotional states to points in a certain space. Different emotional states are distributed in different positions in space according to their different dimensions; the distances between locations reflect the differences between the different emotional states. In the previous research, the valence-arousal, two-dimensional emotion model proposed by Russell [6] in 1980 is the most widely used. The model divides emotion into two dimensions: a valence dimension and an arousal dimension. The negative half of the valence dimension axis indicates negative emotion, while the positive half indicates positive emotion. The largest difference between this model and the discrete emotion model is that the dimensional emotion model is continuous; thus, it has the advantage of being able to express emotion within a wide range and can be used to describe the process of emotion evolution.

The aBCI system has achieved great progress in paradigm design, brain signal processing algorithms and applications. However, these BCI systems still face some challenges. On one hand, it is difficult to fully reflect emotional state based on a single modality because emotion information is easily affected by various noises. On the other hand, some modality is easy to disguise, and it is difficult to reflect the true emotional state, e.g., detecting an accurate expression does not always represent a person’s true emotional state, because a person can disguise an expression. Automated emotion recognition is still impractical in many applications, especially for patients with affective disorders.

From a human performance perspective, emotional expression is mainly divided into the neurophysiological level and the external behavior level. By collecting and analyzing behavioral data and brain image data, we can reveal the relationship between behavior and neural activity and construct theoretical models that map emotion, behavior, and the brain. Therefore, emotion recognition can be based on images, voice, text and other easily collected physiological signals (e.g., skin impedance, heart rate and blood pressure). Nonetheless, it is difficult to use these signals to accurately identify complex emotional states. Among all the types of signals that can be useful in identifying emotional states, electroencephalogram (EEG) data have been widely used because they provide a noninvasive and intuitive method for measuring emotion. EEGs data are the preferred approach for studying the brain’s response to emotional stimuli [7]. EEGs can record neurophysiological signals and they objectively reflect the activity of the cerebral cortex to a certain extent. Studies have shown that different brain regions participate in different perceptual and cognitive activities; for example, the frontal lobe is related to thinking and consciousness, while the temporal lobe is associated with processing complex stimulus information, such as faces, scenes, smells and sounds. The parietal lobe is associated with the integration of a variety of sensory information and the operational control of objects, and the occipital lobe is related to vision [8].

We assume that multimodal emotion recognition based on EEG should integrate not only EEG signals, an objective method of emotion measurement, but also a variety of peripheral physiological signals or behaviors. Compared to single patterns, multimodal emotion processing can achieve more reliable results by extracting additional information; consequently, it has attracted increasing attention. Li et al. [9] proposed that in addition to combining different input signals, emotion recognition should include a variety of heterogeneous sensory stimuli (such as audio-visual stimulation) to induce emotions. Many studies [10,11] have shown that integrating heterogeneous sensory stimuli can enhance brain patterns and further improve brain–computer interface performance.

To date, a considerable number of studies on aBCI have been published, especially on emotion recognition based on EEG signals. In particular, with the development of modality fusion theory and portable EEG devices, many new technologies and applications of emotion recognition have emerged. However, there are few comprehensive summaries and discussions of multimodal EEG-based emotion recognition systems. With this problem in mind, this paper presents the concepts and applications of multimodal emotion recognition to the aBCI community. The main contributions of this paper are as follows:

Therefore, in this study, we present the concepts and applications of multimodal emotion recognition to the aBCI community clearly. The main contributions of this study are as follows:(1)We update and extend the definition of multimodal aBCI. Three main types of multimodal aBCI were devised: aBCI based on a combination of behavior and brain signals, aBCI based on various hybrid neurophysiology modalities and aBCI based on heterogeneous sensory stimuli;(2)For each type of aBCI, we have reviewed several representative multimodal aBCI systems and analyze the main components of each system, including design principles, stimuli paradigms, fusion methods, experimental results and relative advantages. We find that emotion-recognition models based on multimodal BCI achieve better performances than do models based on single-modality BCI;(3)Finally, we identify the current challenges in the academic research and engineering application of emotion recognition and provide some potential solutions.

The remainder of this paper is organized as follows: Section 2 briefly introduces the research basis of emotion recognition and the overall situation of related academic research. Section 3 focuses on data analysis and fusion methods used for EEG and behavior modalities, while Section 4 describes the data analysis and fusion methods used for EEG and neurophysiology modalities. Section 5 summarizes the aBCI based on heterogeneous sensory stimuli. Finally, we present concluding remarks in Section 6 and provide several challenges and opportunities for the design of multimodal aBCI techniques.

## 2. Overview of Multimodal Emotion Recognition Based on BCI

### 2.1. Multimodal Affective BCI

Single modality information is easily affected by various types of noise, which makes it difficult to capture emotional states. D’mello [12] used statistical methods to compare the accuracy of single-modality and multimodal emotion recognition using a variety of algorithms and different datasets. The best multimodal emotion-recognition system reached an accuracy of 85% and was considerably more accurate than the optimal single-modality correspondence system, with an average improvement of 9.83% (the median was 6.60%). A comprehensive analysis of multiple signals and their interdependence can be used to construct a model that more accurately reflects the potential nature of human emotional expression. Different from D’mello [12], Poria [13] conducted a horizontal comparison between multimodal fusion emotion recognition and single-modality emotion recognition on the same dataset with regard to accuracy based on a full discussion of the current situation of single-modality recognition methods. This study also strongly indicated that efficient modality fusion greatly improves the robustness of emotion-recognition systems.

The aBCI system provides an effective method for researching emotional intelligence and emotional robots. Brain signals are highly correlated with emotions and emotional states. Among the many brain measurement techniques (including EEG and functional magnetic resonance imaging (fMRI)), we believe that multimodal aBCI systems based on EEG signals can greatly improve the results of emotion recognition [14,15].

The signal flow in a multimodal aBCI system is depicted in Figure 1. The workflow generally includes three stages: multimodal signal acquisition, signal processing (including basic modality data processing, signal fusion and decision-making) and emotion reflection control. These stages are described in detail below.

(1)In the signal acquisition and input stage, modality data from combined brain signals and behavior modalities (such as expression and eye movement) or various hybrid neurophysiology modalities (such as functional Near-infrared spectroscopy (fNIRS) and fMRI) induced by heterogeneous sensory stimuli (such as audio-visual stimulation) during interaction with other groups or by certain events are captured by the system’s signal acquisition devices. At this stage, multimodal aBCI can be divided into three main categories: aBCI based on a combination of behavior and brain signals, aBCI based on various hybrid neurophysiology modalities and aBCI based on heterogeneous sensory stimuli. For each type of multimodal aBCI, the principles are further summarized, and several representative aBCI systems are highlighted in Section 3, Section 4 and Section 5.(2)In signal processing, obtaining pure EEG signals containing emotional signals is critical. For neurophysiological signals, the preprocessing process is to denoise and remove artifacts from the collected original signals. For image signals such as expressions, irrelevant information in images is eliminated and information representing emotional state is restored and enhanced. In the stage of modalities fusion, we can use different fusion strategies, including data fusion, feature fusion and decision fusion. In decision-making, emotional state decision output can select spatial tensors [16,17], machine learning [18,19], or deep learning methods [20,21], as the final classification decision-maker.(3)Multimodal aBCI systems output relevant emotional states. If the task involves only emotional recognition, it does not require emotional feedback control; the feedback occurs through a relevant interface (such as diagnosis). However, if the task involves social effects (such as wheelchair control), the user’s state information needs to be included directly in the closed-loop system to automatically modify the behavior of the interface with which the user is interacting. Many aBCI studies are based on investigating and recognizing user’ emotional states. The applied domains for these studies are varied and include such fields as medicine, education, driverless vehicle and entertainment.

### 2.2. The Foundations of Multimodal Emotion Recognition

#### 2.2.1. Multimodal Fusion Method

With the development of multisource heterogeneous information fusion, fusing features from multiple classes of emotional states may be important in emotion recognition. Using different types of signals to support each other and fusing complementary information can effectively improve the final recognition effect [22]. The current mainstream fusion modalities include data-level fusion (sensor-layer fusion), feature-level fusion and decision-level fusion.

Data-level fusion [23]—also known as sensor layer fusion—refers to the direct combination of the most primitive and unprocessed data collected by each sensor to construct a new set of data. Data-level fusion processing primarily involves numeric processing and parameter estimation methods. It includes estimation techniques (linear and nonlinear) and uses various statistical operations to process the data from multiple data sources.

Feature-level fusion [24,25] involves extracting a variety of modality data, constructing the corresponding modality features and splicing the extracted features into a larger feature set that integrates the various modality features. The most common fusion strategy cascades all the modality feature data into feature vectors and then inputs them into an emotion classifier.

Decision-level fusion [26,27] involves determining the credibility of each modality to the target and then coordinating and making joint decisions. Compared with feature-level fusion, decision-level fusion is easier to carry out, but the key is to gauge the importance of each modality to emotion recognition. Fusion strategies adopted at the decision level are based on statistical rules [28] (e.g., sum rules, product rules and maximum-minimum-median rules), enumerated weights [29,30], adaptive enhancement [31,32], Bayesian inference and its generalization theory (Dempster–Shafer theory [33], dynamic Bayesian networks [34]) and fuzzy integrals [35].

#### 2.2.2. The Multimodal Open Database and Its Research on Representativeness

The open database is convenient for objective comparison of algorithms; for this reason, much research has focused on the emotion database. To enhance an intuitive understanding of the EEG-based multimodal emotion data sets, we summarize the characteristics of the popular databases in Table 1.

Current research on EEG-based multimodal emotion recognition uses different induction materials and acquisition equipment and lacks objective evaluation of the algorithms. Therefore, to objectively evaluate the performances of different algorithms, the related studies were compared on the same open databases. Table 2 lists the representative multimodal emotion recognition research based on open datasets. However, to provide readers with a more comprehensive understanding of multimodal emotion research, the case studies in Section 3 and Section 4 are no longer limited to open datasets.

## 3. Combination of Behavior and Brain Signals

The core challenge of multimodal emotion recognition is to model the internal working of modalities and their interactions. Emotion is usually expressed through the interaction between neurophysiology and behavior; thus, it is crucial to capture the relationship between them accurately and make full use of the related information.

Basic emotion theory has shown that when emotions are aroused, a variety of human neurophysiological and external behavioral response systems are activated [47]. Therefore, theoretically, the combination of a subject’s internal cognitive state and external subconscious behavior should greatly improve the recognition performance. Concomitantly, the development and promotion of neurophysiological and behavioral signal acquisition equipment has made emotion recognition using mixed multimodalities of human neurophysiological and behavioral performances a research hotspot in international emotional computing. Therefore, mixed modalities involving physiological and external behavior modalities have attracted more attention from emotion recognition researchers.

Among the major modalities of behavior, eye movement tracking signals and facial expressions are two modalities with content-aware information. Facial expression is the most direct method of emotional representation. By analyzing information from facial expressions and combining the results with a priori knowledge of emotional information (Figure 2a), we can infer a person’s emotional state from facial information. Eye movement tracking signals can provide a variety of eye movement indicators and eye movement tracking signals can reflect user subconscious behaviors and provide important clues to the context of the subjects’ current activity [48].

### 3.1. EEG and Eye Movement

Eye-movement signals allow us to determine what is attracting a user’s attention and observe their subconscious behaviors. Eye movements form important cues for a context-aware environment that contains complementary information for emotion recognition. Eye movement signals can provide a rich signal that reflects emotional characteristics, including three common basic features, pupil diameter, gaze information and saccade signals—as well as extended statistical features, including statistical frequency events and statistical deviation events. As shown in Figure 2b, the pupil refers to a resizable opening in the iris of the eye. Related studies [50,51] have shown that pupil size is related to cognition. When we are stimulated from a resting state to an emotional state, our pupils will change size accordingly. Fixation refers to a relatively stable position of the eyeball within a certain range of time and space offset thresholds. Saccade refers to rapid eye movement between two points of interest (or attention). Using these basic characteristics of eye movement signals, we can detect people’s arousal and valence from the statistical values of fixation information and scan signals and then use them to evaluate people’s inner activity states.

Soleymani [19] used two modalities, EEG signals and eye movement data. For EEG data, the unwanted artifacts, trend and noise were reduced prior to extracting the features from EEG data by preprocessing the signals. Drift and noise reduction were done by applying a 4–45 Hz band-pass filter. For eye movement data, a linear interpolation was used to replace the missing pupil diameter samples due to eye blinking. After removing the linear trend, the power spectrum of the pupil diameter variation was computed. In terms of modalities fusion, they applied two fusion strategies: feature-level fusion and decision-level fusion. Feature-level fusion did not improve the best experimental results of EEG and eye gaze data. However, the channel fusion strategy using decision-level fusion increased the optimal classification rates of arousal and valence to 76.4% and 68.5%, respectively. In feature level fusion, feature vectors from different modalities are combined to form a larger feature vector. Then, feature selection and classification methods are applied to the new feature set. Compared with feature-level fusion, the decision-level fusion strategy is more adaptive and scalable, and it allows modalities to be removed or added to the system without using decision-level fusion to retrain the classifier. A support vector machine (SVM) classifier with an RBF kernel is used in both modalities, and the classification results are fused to obtain multimodal fusion results. The authors drew lessons from the confidence and fusion mentioned in the study [8] to complete the fusion of the two modality classification results. The probability output of the classifier is used as a measure of confidence. For a given experiment, the sum rule is defined as follows:(1)$$g_{a}=\frac{\sum_{q\in Q}P_{q}\left(\omega_{a}|x_{i}\right)}{\sum_{a=1}^{K}\sum_{q\in Q}P_{q}\left(\omega_{a}|x_{i}\right)}=\sum_{q\in Q}\frac{1}{\left|Q\right|}P_{q}\left(\omega_{a}|x_{i}\right)$$
where $g_{a}$ is the summed confidence interval for an affected class $\omega_{a}$. Q is the classifier ensemble selected for fusion, $\left|Q\right|$ is the number of classifiers, $P_{q}\left(\omega_{a}|x_{i}\right)$ is the posterior probability of having class $\omega_{a}$ and the sample is $x_{i}$ according to classifier q. To obtain $P_{q}\left(\omega_{a}|x_{i}\right)$, the authors used the MATLAB libSVM implementation [52] of the Platt and Wu algorithms. The final choice was made by selecting the class $\omega_{a}$ with the highest $g_{a}$. Note that $g_{a}$ can also be viewed as a confidence assessment on the class $\omega_{a}$ provided by the fusion of classifiers.

In [37], a multimodal framework called EmotionMeter was proposed to recognize human emotions using six EEG electrodes and eye-tracking glasses. The experiments in this study revealed the complementarity of EEG and eye movement in emotion recognition and a multimodal depth neural network was used to improve the recognition performance. For EEG data, in the preprocessing stage, a band-pass filter between 1 and 75 Hz was applied to filter the unrelated artifacts. Then they used the short-time Fourier transform without overlap in 4 s time window to extract the characteristic power spectral density (PSD) and differential entropy (DE) of five frequency bands of each channel. For eye movements, they extracted various features from different detailed parameters used in the literature, such as pupil diameter, fixation, saccade and blink. Among all the characteristics of eye movements, the pupil diameter is most directly related to the emotional state, but it is easily affected by the ambient brightness [53]. Therefore, it is particularly important to preprocess the influence of light and shadow removal on pupil diameter. Based on the observations that the changes in the pupil responses of different participants to the same stimuli have similar patterns, we applied a principal component analysis (PCA)-based method to estimate the pupillary light reflex. Suppose that Y=A+B+C where Y is original sample data of pupil diameters, A is luminance influences that are prominent, B is emotional influences that we want and C is noise. They used PCA to decompose Y and computed the first principle component as the estimate of the light reflex. To enhance the recognition performance, they adopted a bimodal deep autoencoder (BDAE) to extract the shared representations of both EEG and eye movements. The authors construct two restricted Boltzmann machines (RBMs) for EEG and eye movement data, respectively. They connected the two hidden layers of the EEG-RBM and the eyes-RBM, trained the stacked RBMs and input their results into the BDAE, which then learned a shared representation of the two modalities from the RBMs. Finally, this new shared representation was used as the input to train a linear SVM. Their experiments show that compared with a single modality (eye movement: 67.82%, EEG: 70.33%), multimodal deep learning from combined EEG and eye movement features can significantly improve the accuracy of emotion recognition (85.11%).

Wu et al. [54] applied deep canonical correlation analysis (DCCA) to establish a multimodal emotion-recognition model in which the connecting functional features from EEG and those features from eye movements are fused. The architecture of the DCCA model includes three parts: stacked nonlinear layers to achieve nonlinear transformations for two modalities separately, followed by a canonical correlation analysis (CCA) calculation to maximize the correlation between the two transformed modalities. Finally, the two transformed features are fused by weighted average. An SVM is trained on the fused multimodal features to construct the affective model. Notably, the emotion-relevant critical subnetwork actually lies in the network topology. The three high-dimensional topological features, strength, clustering coefficient and eigenvector centrality, are extracted to represent the connected functional features of EEG. For the eye movement signals, the artifacts were eliminated using signals recorded from the electrooculogram (EOG) and FPz channels. Similar to [37], principal component analysis was adopted to eliminate the luminance reflex of the pupil and to preserve the emotion-relevant components. Next, the BeGaze2 [55] analysis software of SMI (SeoMotoric Itruments, Berlin, Germany) eye movement glasses was used to calculate the eye movement parameters, including pupil diameter, fixation time, blinking time, saccade time and event statistics. Then, the statistics of these parameters were derived as 33-dimensional eye movement features. The experimental results not only revealed the complementary representation characteristics between the EEG functional connection network and eye movement data, but also found that the brain functional connectivity networks based on the 18-channel EEG approach are comparable with that of 62-channel EEG under a multimodal pattern, which provided a solid theoretical basis for the use of portable BCI systems in real situations.

We believe that a highly robust emotion-recognition model can be achieved by combining internal physiological EEG signals and external eye movement behavioral signals. The strategies of the model include extracting discriminative EEG representations and eye movements for different emotions and then combining the EEG and eye movement signals through efficient model fusion methods.

### 3.2. EEG and Facial Expressions

Facial expressions are the most common emotional features. Facial expressions can usually be divided into basic expressions (such as happy and sad), complex expressions, abnormal expressions and microexpressions. The most commonly used emotion recognition methods for facial expressions are geometric and texture feature recognition and facial action unit (facial muscle action combination) recognition.

The texture operator has strong abilities for image-feature extraction. The extracted texture features are mainly used to describe the local gray changes in the image. The representative methods are the Gabor wavelet [56] and local binary pattern [57]. Geometric feature extraction is a classical emotional feature extraction method that typically locates the key points of the human face, measures the relative distances between the positioning points, and finally, defines the features according to the distances. In [58], the authors located eight key points in the eyebrows and the corners of the eyes and mouth. They calculated and obtained six normalized geometric feature vectors and used them in experiments to demonstrate the validity of the geometric feature extraction method.

According to anatomic principles, when a person’s emotional state changes, the facial muscles will be subject to a certain degree of tension or relaxation. This process is dynamic, not static. Various (dynamic) facial expressions can be decomposed into combinations of facial action units (AU), which can be used to recognize complex expressions, abnormal expressions and micro expressions. Ekman et al. developed a facial action coding system (FACS) [59]. Facial expressions can be regarded as combinations of the different facial motion units defined by FACS.

Although emotion recognition methods based on expression are more direct and the facial modality data are easy to obtain, this approach lacks credibility. For example, people may show a smile, but it may be a wry smile. Moreover, it is difficult to detect emotional states through the facial expressions of patients with facial paralysis. Therefore, a combination of facial expression and EEG, which reflect the activities of the central nervous system, will undoubtedly detect changes in people’s emotional states more objectively and accurately.

Huang et al. [60] proposed two multimodal fusion methods between the brain and peripheral signals for emotion recognition. Their approach achieved accuracy rates of 81.25% and 82.75% on four emotion-state categories (happiness, neutral, sadness and fear). Both models achieved accuracies higher than those of either facial expression (74.38%) or EEG detection (66.88%) alone. They applied principal component analysis (PCA) to analyze the facial expression data and extract high-level features and a fast Fourier transform to extract various PSD features from the raw EEG signals. Due to the limited data quantity, they proposed a method that can help prevent overfitting. From the modality fusion aspect, they adopted a decision strategy based on production rules. Such approaches are commonly found in simple expert systems in cognitive modeling and artificial intelligence fields.

However, their study proved that multimodal fusion detection achieves significant improvements compared with single-modality detection methods. Indeed, it may very well reflect how humans conduct emotion recognition: for example, an expression of weak happiness is typically answered with a neutral emotion, whereas a strong expression of sadness usually evokes fear.

In an extension of their work [22], they applied a pretrained, multitask convolutional neural network (CNN) model to automatically extract facial features and detect the valence and arousal values using a single-modal framework. For EEG, they first used the wavelet transform to capture the time-domain characteristics of the EEG when extracting PSD features; then, they used two different SVM models to recognize the values of valence and arousal. Finally, they obtained the decision-level fusion parameters based on training data for both fusion methods. In addition to implementing the widely used fusion method based on enumerating different weights between two models, they also explored a novel fusion method that requires a boosting technique. They employed both classifiers (facial expression and EEG) as subclassifiers of the AdaBoost algorithm to train the weights of the AdaBoost algorithm. The final results were calculated using Equations (2) and (3):(2)$$S_{\mathrm{boost}}=1/(1+\exp(-\overset{n}{\underset{j=1}{\sum }}w_{j}s_{j}))$$
(3)$$r_{\mathrm{boost}}=\left\{\begin{array}{ll} \mathrm{high}\; & S_{\mathrm{boost}\;}\geq 0.5\\ \mathrm{low}\; & S_{\mathrm{boost}\;}<0.5 \end{array}\right.$$
where $S_{\mathrm{boost}}$ stands for the score of AdaBoost score, and the scores of the two classifiers are fused by the AdaBoost method to obtain the final mood score. $r_{\mathrm{boost}}$ represents the prediction results (high or low) of the AdaBoost fusion classifier. $s_{j}\in \left\{-1,1\right\}\left(j=1,2,\ldots ,n\right)$ represents the corresponding output result of j-th sub-classifier. In this case, $S_{1}$ is the facial expression classifier and $S_{2}$ is the EEG classifier. To obtain $w_{j}\left(j=1,2,\ldots ,n\right)$ from a training set of size $m,s{\left(x_{i}\right)}_{j}\in \left\{-1,1\right\}$ designates the output of the j-th classifier for the i-th sample and $y_{i}$ denotes the true label of the i-th sample. They applied this method separately for the online experiment and achieved accuracies of approximately 68.00% for the valence space and 70.00% for the arousal space after fusion—both of which surpassed the highest-performing single-modality model.

To improve the classification accuracy and reduce noise, Sokolov et al. [61] proposed a combined classification method with decision-level fusion for automated estimation of emotion from EEG and facial expressions. In their study, different Hjorth parameters were adopted for different prime emotional quantities (e.g., arousal, valence and dominance) estimation tasks from a given EEG channel and frequency band. For face detection, to reduce the detection error and false alarm rate and speed up the image processing, a cascaded classifier and a monolithic classifier were combined into a two-level classifiers cascade. In the first level, the Haar-like feature cascade of the weak classifier was used to detect face-like objects, while the second level was used for face verification by a convolutional neural network. Then, the face feature was represented by PCA, which preserved the greatest amount of energy with the fewest principal components. They applied an SVM as a classifier for each modality. Finally, the two modalities were combined by the multi-sample–multi-source strategy [20], in which the final output is the average score from both modalities:(4)$$y_{\mathrm{combined}\;}=\frac{1}{M}\overset{M}{\underset{j=1}{\sum }}\left[\frac{1}{N}\overset{N}{\underset{i=1}{\sum }}\left(y_{i,j}\right)\right]$$
where $y_{\mathrm{combined}}$ is the combined output score, M is the number of modalities used (EEG and face) and N is the number of samples for each biometric modality. $y_{i,j}$ is the output of the system for the i-th sample from j-th modality and $y_{i,j}\in \left[0,\ldots ,1\right]$.

Different from eye movement, facial expressions can be captured conveniently by a camera. Therefore, the application scenarios of EEG and facial expression modality fusion are more extensive. However, if subjects feign their facial expressions, they can always successfully “cheat” the machine. Therefore, how to use EEG as auxiliary information to avoid errors caused by subjects camouflaging their true emotions is an important research direction.

## 4. Various Hybrid Neurophysiology Modalities

Emotion usually refers to a state of mind that occurs spontaneously rather than consciously and is often accompanied by physiological changes in the central nervous system and the periphery, affecting EEG signals and heart rate. Many efforts have been made to reveal the relationships between explicit neurophysiology modalities and implicit psychological feelings.

### 4.1. EEG and Peripheral Physiology

In general, physiological reactions are the results of nonautonomic nerves. The physiological reactions and the corresponding signals are difficult to control when emotions are excited. Studies have used such physiological reactions to determine and classify different kinds of emotions [62]. A large number of studies [63,64] have shown that all kinds of physiological signals that change over time have various common characteristics. On one hand, their processing operations are very similar; on the other hand, they can be used to extract features from three dimensions (Figure 3a): the time, frequency and time–frequency domains.

Feature extraction is an important step in emotion evaluation because high-resolution features are critical for effective pattern recognition. The common time domain features include statistical features [65] and Hjorth features [66]. The common frequency domain features include PSD [67], DE [68] and rational asymmetric (RASM) [69] features. The common features in the time–frequency domain include wavelet [70], short time Fourier transform [71] and Hilbert–Huang transform [72] features.

From the aspect of emotion recognition based on various hybrid neurophysiology modalities, a method of emotion recognition of multimodal physiological signals based on a convolutional recurrent neural network is proposed [73]. In this method, a CNN was used to learn the fine spatial representation of multichannel EEG signals and an LSTM was used to learn the time representation of physiological signals. The two features were then combined to perform emotion recognition and classification and tested on the DEAP dataset. In the arousal emotion dimension, the average correct emotion recognition rate was 93.06%; in the valence emotion dimension, the average correct emotion recognition rate was 91.95%. To extract the spatial signals of EEG, this study develops a position mapping and uses “0” to represent the unused electrode channels in the DEAP dataset. The international 10–20 system is generalized with the test electrodes used in the DEAP dataset to a matrix (h × w), where h is the maximum number of vertical test points and w is the maximum number of horizontal test points. Through position mapping, a one-dimensional EEG data vector sequence is transformed into a two-dimensional EEG data matrix sequence. Finally, the two-dimensional matrix sequence is divided into several groups of two-dimensional matrix sequences. Each matrix sequence has a fixed dimension, and no overlap occurs between two consecutive matrix sequences. To extract the temporal feature of physiological modalities, the signals collected from 33–40 channels of the DEAP dataset are two EOG, two EMG, one GSR, one RSP, one BVP and one ST. Two stacked recurrent neural network (RNN) layers are built. Each layer of the RNN includes multiple LSTM units, and the output of the first RNN layer forms the input to the second RNN. At the fusion level, they use feature-level fusion to combine the extracted spatial features from the EEG signals with the temporal features extracted from the physiological signals. These fused features are input to a softmax layer to predict the emotional state. The final results are calculated using Equation (5):(5)$$P_{j}=\mathrm{softmax}\left(\left[V_{S},V_{T}\right]\right),P_{j}\in R^{2}$$
where $V_{S}$ represents the spatial feature vector of EEG, $V_{T}$ is the temporal feature vector of the physiological signals and $P_{j}$ stands for the predicted emotional state.

Kwon et al. [74] proposed improving emotion classification accuracy and reducing model instabilities using a CNN model based on EEG and GSR data. First, a wavelet transform was applied to represent the EEG signal in the frequency domain. The EEG signal was collected from 32 electrodes and derived from a single stimulus. To learn the EEG features from all channels in the spectrogram image, this study adopted a multilayer 2D CNN instead of the one-layer, one-dimensional CNN applied in prior conventional research. In this way, the network was able to extract efficient temporal–spatial features for EEG. The GSR signals are preprocessed by using the short time zero-crossing rate (STZCR). After passing through the various layers of the CNN, a nonlinear activation function and a pooling layer, the EEG image is flattened and combined with the STZCR of the GSR. Based on these fused features, a fully connected layer and softmax were used to perform classification. The author applied data processing to denoise and design a reasonable network architecture to extract efficient features and combine features, and finally, was able to improve the classification performance significantly.

Currently, the advanced algorithms of multimodal emotion recognition based on EEG and physiological signals requires considerable training data to construct a high-quality machine learning model. This computational complexity limits the application of these advanced algorithms in portable devices and scenarios requiring timeliness.

Hyperdimensional (HD) computing [75] has demonstrated rapid learning ability on various biological signal processing tasks [76,77], each of which operates with a specific type of biological signal (see the overview in [78]). Chang et al. [79] extended single-task HD processing to multitask processing and applied it to physiological signals from multimodal sensors. The experimental results on the AMIGOS datasets show that the method achieved a balance between accurate and necessary training data to a certain extent; the proposed model’s average classification accuracy was the highest, reaching 76.6%, and its learning speed was the fastest. They also proposed multimodal emotion recognition (HDC-MER) based on high-definition computing. HDC-MER uses random nonlinear functions to map real-valued features to binary HD vectors, further encodes them over time and fuses various patterns, including GSR, ECG and EEG. HDC-MER includes two stages: in the first stage, the original features are transformed into high-definition binary embedded features and in the second stage, multimodal fusion, learning and classification are performed. First, the spatial encoder bundles all the modality feature information at a given point in time by using most of the functions of the HD vector components. A time encoder is used to collect feature changes between videos to capture time-dependent mood swings. After spatial and time domain coding is completed, the next step is to fuse multimodal HD vectors. The fusion unit bundles the corresponding HD vectors into a fused d-bit HD vector. The output from the fusion unit is sent to the associative memory for training and inference. During the training phase, the fusion unit generated from the training data is bundled to its corresponding class prototype of the associative memory. That is, associative memory collects the fusion unit of the same class and bundles them to a prototype HD vector by the majority function. During the inference phase, they used the same encoding, but the label of the fusion unit is unknown; hence, they call it the query HD vector. To perform classification, the query HD vector is compared with all learned prototype HD vectors to identify its source class according to the Hamming distance (i.e., a measure of similarity), defined as the number of bits that differ between the two HD vectors. Finally, two emotional labels with the minimum distance are returned.

Based on the combination of peripheral physiological signals and central nerve signals, this approach can avoid recognition errors caused by emotional camouflage, and patients with dyskinesia can also be effectively processed.

### 4.2. EEG and Other Neuroimaging Modality

Although the arousal index of human emotional activity can be effectively detected to a certain extent by measuring the activity of the human peripheral nervous system, this approach cannot effectively measure the central nervous system of the human body. As a classic cognitive style, the generation of positive emotion mainly depends on related brain activities. To date, functional neuroimaging has become a more direct tool for exploring and explaining brain activity (Figure 3b). The low spatial resolution of EEG is one of its shortcomings in the study of emotional activity. In addition to EEG based on neural electrical activity, brain imaging techniques based on blood oxygen imaging, such as fNIRS and fMRI, have been widely used in neuroimaging research. Therefore, in the application of emotion recognition, integrating various brain imaging modality information can provide high-resolution spatiotemporal neural images that can help in further understanding the brain rules when emotional states occur. Then, related emotions can be identified.

Compared with EEG, the advantage of near-infrared technology is that it is not affected by widespread environmental electrical noise, and its sensitivity to EMG artifacts is much lower than that of EEG. Near-infrared light is absorbed when it penetrates biologic tissue; the measurement results of near-infrared spectroscopy are related to brain activity and are attributed to the effect of this interaction. The slow hemodynamic response showed that Hb increased slightly after the beginning of the neural activity, followed by a large, but delayed increase in $HbO_{2}$, which reached a peak at approximately 10 s [80,81] after activation, while Hb decreased correspondingly [82]. The changes in the concentration of oxygenated hemoglobin and deoxyhemoglobin can be calculated using the modified Beer-Lambert law (shown in Equation (6)) based on the detected changes in light intensity [83].
(6)$$\left[\begin{array}{c} \Delta Hb\\ \Delta HbO_{2} \end{array}\right]={\left[\begin{array}{cc} \alpha_{deaxy\left(\lambda 1\right)} & \alpha_{oxy\left(\lambda 1\right)}\\ \alpha_{deaxy\left(\lambda 2\right)} & \alpha_{axy\left(\lambda 2\right)} \end{array}\right]}^{-1}\cdot \left[\begin{array}{c} \Delta A_{\left(\lambda 1\right)}\\ \Delta A_{\left(\lambda 2\right)} \end{array}\right]\cdot B$$
where Δ represents the amount of change at a given time relative to an initial time, and α indicates the absorption rate of a certain hemoglobin (oxy or deaxy) to a certain wavelength of light (λ1 or λ2). *A* represents the light intensity of a certain wavelength detected. B indicates the length of the optical path, which is related to the distance between the emitters and receivers (usually 3 cm) and is a constant predetermined by the experiment. Thus, the sum of two unknowns (ΔHb and $\Delta HbO_{2}$) can finally be calculated. However, the fNIRS system measures the hemodynamic response, which takes several seconds to develop. The delay in the hemodynamic response has been estimated by modeling simulations and computational methods [84,85]. More invasive methods also demonstrate delayed hemodynamic responses [86].

However, fNIRS technology has some problems; for example, the time resolution is insufficient and cannot directly reflect neural activity, which seriously affects the reliability of fNIRS in detecting brain function. However, fNIRS can determine the brain function signal source through channel localization. In contrast, EEG, which is a mature brain function detection technology, has high time resolution, but its disadvantage is that the detection signal cannot identify the brain region source. On the other hand, both fNIRS and EEG techniques have the advantages of small restrictions on the environment and subjects and can be used to detect the brain functions of subjects in a state of natural relaxation. Therefore, combining fNIRS and EEG technology into a bimodal detection technology capitalizes on their spatiotemporal resolution advantages and can help people better understand the neural mechanisms of brain activity in cognitive psychological tasks.

A multimodal method for the joint evaluation of fNIRS and EEG signals for emotional state detection was proposed in [87]. The emotional state was recorded from video capture of facial expressions, and the related neural activity was measured by wearable and portable neuroimaging systems: functional fNIRS and EEG, which can evaluate hemodynamic and electrophysiological responses, respectively. EmotivEPOC (Neurotech, San Francisco, CA, USA) headphones were used to collect EEG signals at a 128 Hz sampling rate, while fNIRS devices and sampling wireless fNIRS systems were used to monitor the prefrontal cortex of the receptors to avoid the problem of overlapping brain regions. The method included simultaneous detection and comparison of various emotional expressions through multimodalities and classification of spatiotemporal data with neural characteristics. The experimental results showed a strong correlation between spontaneous facial emotion expression and brain activity related to an emotional state. By comparing the recognition results of the system with the actual tags of the test images, the recognition accuracy of the system was approximately 74%. At the same time, the experimental results showed that the method using fNIRS and EEG achieved better performance than did the method using only fNIRS or EEG. Nevertheless, this study did not explain in detail the time synchronization method for the different time resolutions and measurement delays between fNIRS and EEG.

In practical applications of BCI systems, time synchronization may be a key problem because the information transfer rate is the most important factor in evaluating BCI systems. To address these problems, computational methods such as using prior information [88] or normalized features [89] were proposed to obtain better BCI performance than a single modality. Morioka et al. [88] used fNIRS features as prior information to estimate cortical currents in EEG, while Ahn et al. [89] combined EEG and fNIRS features by normalizing all features into the range [0,1] and applied the sum of the features. Although further optimization steps are still needed, these two novel approaches may become future solutions for overcoming the current limitations in integrating EEG-fNIRS features.

To the best of our knowledge, in the field of computer engineering, there are few studies on emotion recognition that integrate EEG-fNIRS, and most of them focus on the task of motion imagination. For example, in [90], the performance of a sensory motor rhythm-based BCI was significantly improved by simultaneously measuring EEG and NIRS [91]. Some studies [80,92] have shown that hemodynamic changes may be a promising indicator to overcome the limitations of command detection. It is expected that more researchers will conduct in-depth research on integrating EEG-fNIRS for emotion recognition in the future.

To monitor brain activity through fMRI, the principle is that the brain contains a large number of hydrogen protons in rich water, and the spins of hydrogen protons emit electromagnetic waves at a certain frequency under the action of an external main magnetic field (B0). If the proper radio frequency (RF) current is used to excite the protons from a direction perpendicular to the main magnetic field, the spin precession angle increases, and when the excitation current is removed, the proton will return to its original state and emit the same signal as the excitation frequency. To take advantage of this phenomenon, a coil is used to receive signals transmitted from the body in vitro for imaging (Figure 3b). Compared with EEG and fNIRS, the greatest advantage of monitoring brain activity with fMRI is that the spatial resolution is higher than fNIRS; nevertheless, like fNIRS, the time resolution is lower than that of EEG, which makes it difficult to achieve real-time data collection. Additionally, the current application environment of fMRI is relatively limited, which also limits its application scenarios. As far as we know, in the field of computer engineering, no research exists on using combined EEG-fMRI for emotion recognition; most of the existing studies focus on the neural mechanism.

## 5. Heterogeneous Sensory Stimuli

In the study of emotion recognition, to better simulate, detect and study emotions, researchers have employed use a variety of materials to induce emotions, such as pictures (visual), sounds (audio) and videos (audio-visual). Many studies [36,37] have shown that video-induced stimulation based on visual and auditory multisensory channels is effective because audio-visual integration enhances brain patterns and further improves the performance of brain–computer interfaces. Vision and audition, as the most commonly used sensory organs, can effectively provide objective stimulation to the brain, while the induction ability of other sensory channels, such as olfaction and tactile sensation, is still lacking. As the most primitive sense of human beings, olfaction plays an important role in brain development and the evolution of human survival. Therefore, in theory, it can provide more effective stimulation than can vision and audition.

### 5.1. Audio-Visual Emotion Recognition

Multisensory channel integration to induce subjects’ emotions, especially video material that involves emotion recognition evoked by visual and auditory stimuli, will be a future trend. For audio-visual emotion recognition, a real-time BCI system to identify the emotions of patients with consciousness disorders was proposed [10]. Specifically, two classes of video clips were used to induce positive and negative emotions sequentially in the subjects, and the EEG data were collected and processed simultaneously. Finally, instant feedback was provided after each clip. Initially, they recruited ten healthy subjects to participate in the experiment, and their BCI system achieved a high average online accuracy of 91.5 ± 6.34%, which demonstrated that the subjects’ emotions had been sufficiently evoked and efficiently recognized. Furthermore, they applied the system to patients with disorders of consciousness (DOC), who suffer from motor impairment and generally cannot provide adequate emotional expressions. Thus, using this BCI system, doctors can detect the potential emotional states of these patients. Eight DOC patients participated in the experiment and significant online accuracy achieved for three patients. These experimental results indicate that BCI systems based on audio-visual stimulation may be a promising tool for detecting the emotional states of patients with DOC.

### 5.2. Visual-Olfactory Emotion Recognition

Although the affective information from audio–visual stimuli has been extensively studied, for some patients with impaired visual and auditory abilities, such integration cannot play a role. If multisensory channels could be combined with odor to induce emotion stimulation, new discoveries and breakthroughs may be achieved in emotion recognition research, especially for those patients with normal olfactory function.

However, due to the volatile gases of odorant mixtures, the influences of peripheral organs, and the role of the brain, confusion can easily occur in these processes, and it is difficult to quantify. Therefore, research on odor-induced emotions is relatively rare. The representative example of multiple sensory emotion recognition related to olfactory activity is a single study that induced emotion in a patient with DOC [93]. During the experiment, the patient was asked to imagine an unpleasant odor or to ‘relax’ in response to either a downward pointing arrow or a cross appearing on a screen, respectively. The patients’ electrophysiological responses to stimuli were investigated by means of EEG equipment and analyzed using a specific threshold algorithm. A significant result was observed and calculated, which shows that this paradigm may be useful for detecting covert signs of consciousness, especially when patients are precluded from carrying out more complex cognitive tasks.

## 6. Open Challenges and Opportunities

In this section, we discuss the important open issues, which may become popular in research on multimodal emotion recognition based on EEG in the future. In research of multimodal emotion recognition based on EEG, we found many technical challenges and opportunities (Figure 4). Here, we focus on some important research opportunities, which may also be obstacles to aBCI leaving the laboratory and moving to practical application.

### 6.1. Paradigm Design

For general BCI paradigms, the approaches of neurophysiology-informed affect sensing can be categorized in terms of their dependence on user volition and stimulation [3]. The user initiative means that the user sends instructions to the BCI system by actively thinking. The degree of stimulation depends on the users and essentially refers to whether the stimulation is specific (e.g., whether it is necessary to implement different stimuli for different users). This section introduces the latest emotion-related experimental paradigm according to the classification scheme presented in [94] and provides corresponding application suggestions.

#### 6.1.1. Stimulus-Independent and Passive Paradigms

To explore emotions, researchers use general materials related to different emotions to design emotional paradigms, such as the international affective pictures (IAPS) [95], music, movie clips and videos. Pan [96] proposed an experimental paradigm based on open pictures. A facial expression was displayed in the center of the monitor. The emotional content of these images was measured by a self-assessment model (SAM) [97] that contained nine valence and arousal dimensions. The presentation time for each picture was eight seconds. During the presentation, the subjects were asked to focus on the smiling or crying faces. At the completion of the experiment, each subject was asked to mark each picture with the SAM. A music-induced method [98] can spontaneously lead subjects to a real emotional state. In [99], the researchers asked subjects to evoke emotions by recalling an unpleasant smell. Many studies [100,101] have shown that when people receive both auditory and visual sensory inputs, their brains may integrate the auditory and visual features of these stimuli, and audio–visual integration may be accompanied by increased brain activity and state. Huang et al. [22] developed a real-time BCI system based on audio-visual channels to identify emotions evoked by video clips. The researchers used video clips representing different emotional states to test and train subjects.

In this paradigm, the subjects generally self-report the emotional experience scale; thus, it is difficult to know whether the subjects truly received the passive stimulus during the experiment, but because this experimental design is relatively simple and does not require any elaborate laboratory settings, it is widely used.

#### 6.1.2. Stimulus-Independent and Active Paradigms

In the design of this paradigm, some universal participatory materials can be used while simultaneously encouraging subjects to actively achieve a specific target state in some way during the experiment. The most common paradigm is to design emotions by having subjects participate in games. When players realize that their emotional state has an impact on the game parameters, they begin to actively induce their own state to manipulate the game environment according to their preferences [102]. Because the experimental material is universal and the task is completed actively by the user, this paradigm is both expansible and objective.

#### 6.1.3. Stimulus-Dependent and Passive Paradigms

This paradigm is mainly customized based on the user’s characteristics. Specifically, we can selectively provide or automatically play back media items that are known to induce a specific emotional state in certain users. Pan et al. [103] used this paradigm to identify the state of consciousness of patients with consciousness disturbances. In each experiment, the paradigm begins with an audio-visual description in Chinese: “focus on your own photo (or a stranger’s photo) and count the number of frame flash lasting for eight seconds, indicating the target photo. Next, two photos appear, one of which has a flashing frame. The flashing frame is randomly selected and flashes five times. Ten seconds later, one of the two photos identified by the BCI algorithm appears in the center of the graphical user interface (GUI) as feedback. If the result is correct, a scale symbol, a positive audio feedback clip of applause and a detected photo are presented for four seconds to encourage the patient; otherwise, a question mark (‘?’) is presented. Patients are asked to selectively focus on the stimulus related to one of the two photos according to the audio–visual instructions (that is, a voice in the headset and sentences on the screen at the same time).

#### 6.1.4. Stimulus-Dependent and Active Paradigms

Stimulus-dependent and active paradigms are relatively rarely used. Because this approach requires subjects to independently recall the past to stimulate their emotions, it is difficult to effectively control the changes in their emotional states. However, some studies [104,105] have shown that emotional self-induction techniques, such as relaxation, are feasible as control methods.

### 6.2. Modality Measurement

The most important thing for multimodal emotion identification is the measurement and collection of modality data. However, the portability, intrusiveness, cost and integration of the measuring equipment determine the key factors as to whether the multimodal emotion-recognition system can be widely used. The detection equipment for the behavior modality is relatively mature and has reached highly useful levels in terms of portability and the quality of the obtained signals. For face detection, a variety of high-fidelity cameras are very advanced options. For eye movement signal acquisition, conventional eye movement measurement methods include desktop-captured eye movements and glasses-type captured eye movements. Recently, it has been shown that CNNs can be used to extract relevant eye movement features from images captured by smart-phone cameras [106].

For brain imaging equipment, electrodes are directly implanted into the cerebral cortex by surgery, allowing high-quality neural signals to be obtained. However, these procedures pose safety risks and high costs, including wound healing difficulties and inflammatory reactions. In contrast, electroencephalograph leads attach to the scalp and fNIRS avoid expensive and dangerous operations. However, due to the attenuation effect of the skull on the brain signal, the signal strength and resolution obtained are weaker than those of the intrusive acquisition equipment. Although fMRI is noninvasive and has high temporal and spatial resolution, it is expensive and requires users to be in a relatively closed space; consequently, it is difficult to expand to wide use. A team of researchers [107] at the University of Nottingham in the UK has developed a magnetoencephalography system that can be worn like a helmet that allows the scanned person to move freely and naturally during the scanning process. This approach may lead to a new generation of lightweight, wearable neuroimaging tools in the near future, which in turn would promote practical applications of BCI based on brain magnetism.

For physiological signal acquisition, Chen et al. [108] designed “intelligent clothing” that facilitates the unnoticeable collection of various physiological indices of the human body. To provide pervasive intelligence for intelligent clothing systems, a mobile medical cloud platform was constructed by using mobile Internet, cloud computing and big data analysis technology. The signals collected by smart clothing can be used for emotion monitoring and emotion detection, emotional care, disease diagnosis and real-time tactile interaction.

Concerning multimodal integrated acquisition devices, the VR-capable headset LooxidVR produced by LooxidLabs (Daejeon, Korea) integrates head-mounted display (HMD) with built-in EEG sensors and eye tracking sensors. In addition, a phone can be attached to display virtual reality (VR) content [109]. This method can achieve direct synchronization and synchronous acquisition of eye movement tracking and matching EEG data, thus realizing high-fidelity synchronized eye-movement tracking and EEG data to augment VR experiences. The BCI research team [110] of the Technical University of Berlin has released a wireless modular hardware architecture that can simultaneously collect EEG and functional near-infrared brain functional images—as well as other conventional physiological parameters such as ECG, EMG and acceleration. A similar high-precision, portable and scalable hardware architecture for multiphysiological parameter acquisition is a prerequisite for engineering applications for multimodal emotion recognition research. In other words, the improvements in hardware for collecting modality signals will improve the signal quality, widen the user population and promote reform and innovation in the field of aBCI.

### 6.3. Data Validity

#### 6.3.1. EEG Noise Reduction and Artifact Removal

EEG signals have been widely used in medical diagnosis, human-computer interaction, neural mechanism exploration and other research fields. However, EEG signals are extremely weak and easily polluted by unwanted noise, which leads to a variety of artifacts [111]. Artifacts are unwanted signals; these mainly occur stem from environmental noise, experimental errors and physiological artifacts [112]. Environmental artifacts and experimental errors from external factors are classified as external artifacts, while physiological factors from the body (such as blinking, muscle activity and heartbeat) can be classified as inherent artifacts [113,114]. To obtain high-quality EEG signals, in addition to the improvement and promotion of hardware acquisition equipment mentioned, effective preprocessing (noise reduction and artifact removal) of EEG signals is also very important.

For EEG processing, first of all, we should avoid artifacts from the source as much as possible and tell the subjects not to blink or do some actions that may cause artifacts before the start of EEG signal acquisition. Then for some unavoidable artifacts (EOG, EMG, ECG), we need to consider whether the EEG signal and the artifact signal overlap in frequency. If the frequencies do not overlap, we can try to remove the pseudo by linear filtering. For example, the frequency of the five frequency bands (delta: 1–3 Hz, theta: 4–7 Hz; alpha: 8–13 Hz; beta: 14–30 Hz; gamma 31–50 Hz) related to emotion of EEG is 1 ~50 Hz. Low-frequency filter is used to remove EMG artifacts with higher frequency and high-frequency filter is used to remove ocular artifacts with lower frequency. When the artifacts of electrophysiological signals such as frequency overlap, high amplitude and wide frequency band exist, the preprocessing of EEG signals needs to identify and separate the artifacts while retaining the EEG signals containing emotional information. Blind source separation (BSS) [115] techniques are commonly used to remove related artifacts, including canonical correlation analysis (CCA) [116] and independent component analysis (ICA) [117].

#### 6.3.2. EEG Bands and Channels Selecting

The “emotional brain” theory [118] reveals that not every brain region is associated with emotional tasks. This inspires researchers to choose areas of the brain that are closely related to emotion for EEG measurement. At the same time, some studies [119] have shown that there is a close relationship between different EEG bands and different emotional states. By selecting the key frequency bands and channels and reducing the number of electrodes, not only can the computational costs be reduced, but the performance and robustness of the emotion-recognition model can be significantly improved, which is highly important for the development of wearable BCI devices.

Zheng et al. [120] proposed a novel critical channel and frequency band selection method through the weight distributions learned by deep belief networks. Four different configurations of 4, 6, 9 and 12 channels were selected in this experiment. The recognition accuracy of these four configurations is relatively stable, achieving a maximum accuracy of 86.65%—even better than that of the original 62 channels. They also compared the performances of DE characteristics in different frequency bands (delta, theta, alpha, beta and gamma) and found that the performance of the gamma and beta bands was better than that of other frequency bands. These results confirm that the β and γ oscillations of brain activity have a greater relationship with emotional processing than do other frequency oscillations.

#### 6.3.3. Feature Optimization

Many studies [65,121] have found that different feature vectors have different effects on the accuracy of emotional state classification. Combining a variety of different features into high-dimensional feature vectors can also improve the classification accuracy. In addition, different subjects have different sensitivities to different features. Not all features carry important information about the emotional state. Irrelevant and redundant features not only increase the risk of overfitting, but also increase the difficulty of emotion recognition because of increasing the feature space.

Therefore, to eliminate the influence of feature irrelevance and redundancy, improve the real-time performance of the classification algorithm and improve the accuracy of multimodal emotion recognition, feature optimization is usually needed after feature extraction. Feature optimization can be divided into two types: feature selection and feature dimensionality reduction. We can assume that the feature set after feature extraction is $\left\{X_{1},\ldots ,X_{n}\right\}$, the features to be selected are $X_{i}$, the optimized features are $Y_{i}$, the number of features is n and m and the feature selection algorithm, which evaluates and selects the optimal feature subset with strong emotional representation ability from the original set of features is $f_{s}:\left\{X_{1},\ldots ,X_{n}\right\}\to \left\{X_{1},\ldots ,X_{m}\right\}\left(n\leq m\right)$. The feature dimensionality reduction algorithm is $f_{d}:\left\{X_{1},\ldots ,X_{n}\right\}\to \left\{Y_{1},\ldots ,Y_{m}\right\}\left(n\leq m\right)$, which maps the original feature set to a new feature set whose feature attributes are different than those of the original features.

**Feature selection.** Selecting features with a strong ability to represent emotional state is highly important in emotion recognition tasks. Many algorithms exist that can reduce dimensionality by removing redundant or irrelevant features, such as particle swarm optimization (PSO) [122] and the genetic algorithm (GA) [123].**Feature dimension deduction.** The essence of feature dimension reduction is a mapping function that maps high-dimensional data to a low-dimensional space and creates new features through linear or nonlinear transformations of the original eigenvalues. The dimensionality reduction methods are divided into linear and nonlinear dimensionality reduction. Examples of linear dimensionality reduction include principal component analysis (PCA) [124] and independent component analysis (ICA) [125], etc. Nonlinear dimensionality reduction is divided into methods based on kernel functions and methods based on eigenvalues, such as kernel-PCA [126].

### 6.4. Generalization of Model

In practical applications, trained emotional models often need to remain stable for long periods; however, the physiological signals related to emotion often change substantially and data from different individuals will present differently in different time environments. In practical multimodal emotion identification, various problems often occur, such as modality data loss and incomplete modality data collection. Thus, whether we can conduct cross-modality emotion identification also involves system generalizability. However, there are natural differences in the data between different modalities. Generally, these problems limit the popularization and wide application of emotion-recognition systems. The important challenges in achieving a multimodal emotion-recognition system with strong generalizability lie in solving the problems of modality–modality, object–object and session–session variability. Currently, we can adopt the following three types of strategies to solve these problems:(1)A general classifier is trained, but the traditional machine-learning algorithms are based on the assumption that the training data and the test data are independently and identically distributed. Therefore, when the general classifier encounters new domain data, its performance often degrades sharply.(2)Look for effective and stable feature patterns related to emotion. The authors of [69] found that the lateral temporal areas activate more for positive emotions than for negative emotions in the beta and gamma bands, that the neural patterns of neutral emotions have higher alpha responses at parietal and occipital sites and that negative emotional patterns have significantly higher delta responses at parietal and occipital sites and higher gamma responses at prefrontal sites. Their experimental results also indicate that the EEG patterns that remain stable across sessions exhibit consistency among repeated EEG measurements for same participant. However, more stable patterns still need to be explored.(3)For the problem of differences in data distribution, the most mature and effective approach is to use transfer-learning to reduce the differences between different domains as much as possible. Assume that $D_{s}=\left\{X_{S},Y_{S}\right\}$ and $D_{t}=\left\{X_{t},Y_{t}\right\}$ represent the source domain and the target domain, respectively, where X∈x,Y∈y is the sample data and the label. To overcome the three challenges, we need to find a feature transformation function f that makes the edge probability distribution and the conditional probability distribution satisfy $P\left(f\left(X_{s}\right)\right)=P\left(f\left(X_{t}\right)\right)$ and $P(Y_{s}|f\left(X_{s}\right))=P(Y_{t}|f\left(X_{t}\right))$, respectively.For the subject–subject problem, other objects are the source domain, and the new object is the target domain. Due to the large differences in EEG among different subjects, it is traditional to train a model for each subject, but this practice of user dependence is not in line with our original intention and cannot meet the model generalization requirements. In [127], the authors proposed a novel method for personalizing EEG-based affective models with transfer-learning techniques. The affective models are personalized and constructed for a new target subject without any labeled training information. The experimental results demonstrated that their transductive parameter transfer approach significantly outperforms other approaches in terms of accuracy. Transductive parameter transfer [128] can capture the similarity between data distributions by taking advantage of kernel functions and can learn a mapping from the data distributions to the classifier parameters using the regression framework.For the session–session problem, similar to the cross-subject problem, the previous session is the source domain, and the new session is the target domain. A novel domain adaptation method was proposed in [129] for EEG emotion recognition that showed superiority for both cross-session and cross-subject adaptation. It integrates task-invariant features and task-specific features in a unified framework and requires no labeled information in the target domain to accomplish joint distribution adaptation (JDA). The authors compared it with a series of conventional and recent transfer-learning algorithms, and the results demonstrated that the method significantly outperformed other approaches in terms of accuracy. The visualization analysis offers insights into the influence of JDA on the representations.For the modality–modality problem, the complete modality data are the source domain, and the missing modality data are the target domain. The goal is to achieve knowledge transfer between the different modality signals. The authors of [127] proposed a novel semisupervised multiview deep generative framework for multimodal emotion recognition with incomplete data. Under this framework, each modality of the emotional data is treated as one view, and the importance of each modality is inferred automatically by learning a nonuniformly weighted Gaussian mixture posterior approximation for the shared latent variable. The labeled-data-scarcity problem is naturally addressed within our framework by casting the semisupervised classification problem as a specialized missing data imputation task. The incomplete-data problem is elegantly circumvented by treating the missing views as latent variables and integrating them out. The results of experiments carried out on the multiphysiological signal dataset DEAP and the EEG eye-movement SEED-iv dataset confirm the superiority of this framework.

### 6.5. Application

Emotion recognition based on BCI has a wide range of applications that involve all aspects of our daily lives. This section introduce potential applications from two aspects: medical and nonmedical.

#### 6.5.1. Medical Applications

In the medical field, emotion recognition based on BCI can provides a basis for the diagnosis and treatment of mental illnesses. The diagnosis of mental illness has remained at a subjective scale and lacks objective and quantitative indicators that can help clinicians diagnose medical treatments.

Computer-aided evaluation of the emotions of patients with consciousness disturbances can help doctors better diagnose the physical condition and consciousness of patients. The existing research on emotion recognition mainly involves offline analysis. For the first time, Huang et al. [10] applied an aBCI online system to the emotion recognition of patients with disorders of consciousness. Using this system, they were able to successfully induce and detect the emotional characteristics of some patients with consciousness disorders in real time. These experimental results showed that aBCI systems hold substantial promise for detecting emotions of patients with disorders of consciousness.

Depression is a serious mental health disease that has high social costs. Current clinical practice depends almost entirely on self-reporting and clinical opinions; consequently. There is a risk of a series of subjective biases. The authors of [130] used emotional sensing methods to develop diagnostic aids to support clinicians and patients during diagnosis and to help monitoring treatment progress in a timely and easily accessible manner. The experiment was conducted on an age- and sex-matched clinical dataset of 30 patients and 30 healthy controls. The results of experiments showed the effectiveness of this framework in the analysis of depression.

Mood disorders are not the only criteria for diagnosing autism spectrum disorders (ASD). However, clinicians have long relied on the emotional performances of patients as a basis for autism. The results of the study in [131] suggested that cognitive reassessment strategies may be useful for children and adolescents with ASD. Many studies have shown that emotion classification based on EEG signal processing can significantly improve the social integration abilities of patients with neurological diseases such as amyotrophic lateral sclerosis (ALS) or acute Alzheimer’s disease [132].

#### 6.5.2. Non-Medical Applications

In the field of education, students wore portable EEG devices with an emotion-recognition function, allowing teachers to monitor the students’ emotional states during distance instruction. Elatlassi [133] proposed to model student engagement in online environments using real-time biometric measures and using acuity, performance and motivation as dimensions of student engagement. Real-time biometrics are used to model acuity, performance and motivation include EEG and eye-tracking measures. These biometrics have been measured in an experimental setting that simulates an online learning environment.

In the field of driverless vehicles, emotion recognition based on EEG adds an emotion-recognition system to the autopilot system, thus increasing the driving reliability of the automatic driving system [134]. At the same time, a human-machine hybrid intelligent automatic driving system in man’s loop is built. To date, in the automatic driving system, because passengers do not know whether the driverless vehicle can correctly identify and assess the traffic condition or whether it can make the correct judgment and response in the process of driving, passengers are still very worried about the safety of pilotless driving. The brain computer interface technology can detect the passenger’s emotion in real time and transmit the real feeling of the passenger in the driving process to the driverless system, which can adjust the driving mode according to the passenger’s emotional feeling. In the whole system, the human being as a link of an automatic driving system is very good for man-machine cooperation.

In entertainment research and development, we can build a game assistant system for emotional feedback regulation based on EEG and physiological signals that provided players with a full sense of immersion and extremely interactive experiences. In the study of [135], EEG-based “serious” games for concentration training and emotion-enable applications including emotion-based music therapy on the web were proposed and implemented.

## 7. Conclusions

Achieving accurate, real-time detection of user emotional states is the main goal of the current aBCI research efforts. The rapidly expanding field of multimodal sentiment analysis shows great promise in accurately capturing the essence of expressed sentiments. This study provided a review of recent progress in multimodal aBCI research to illustrate how hBCI techniques may be implemented to address these challenges. The definition of multimodal aBCI was updated and extended, and three main types of aBCI were devised. The principles behind each type of multimodal aBCI were summarized, and several representative aBCI systems were highlighted by analyzing their paradigm designs, fusion methods and experimental results. Finally, the future prospects and research directions of multimodal aBCI were discussed. We hope that this survey will function as an academic reference for researchers who are interested in conducting multimodal emotion-recognition research based on combining EEG modalities with other modalities.

## Figures and Tables

**Figure 1 brainsci-10-00687-f001:**
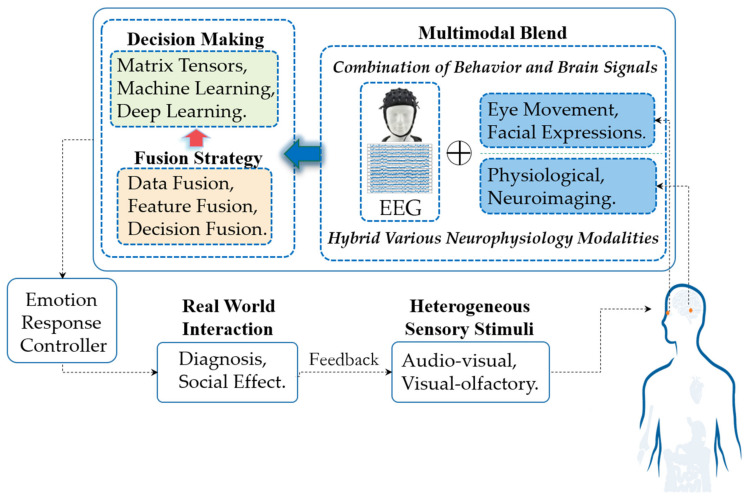
Flowchart of multimodal emotional involvement for electroencephalogram (EEG)-based affective brain–computer interfaces (BCI).

**Figure 2 brainsci-10-00687-f002:**
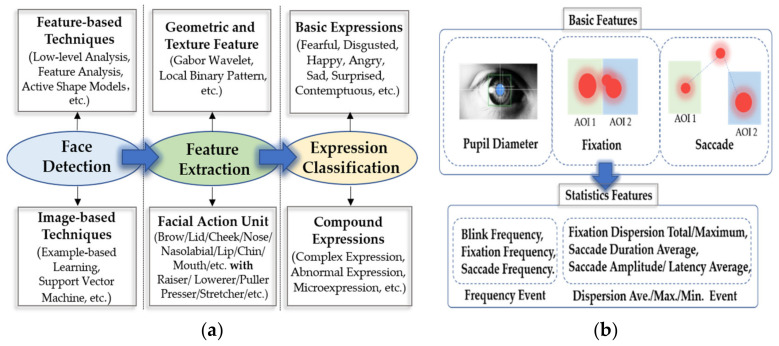
(**a**) Flowchart of emotion recognition based on expression. The facial expression recognition process includes three stages: face location recognition, feature extraction and expression classification. In the face location and recognition part, two technologies are usually adopted: feature-based and image-based [49]. The most commonly used emotion recognition methods for facial expressions are geometric and texture feature recognition and facial action unit recognition. Expressions are usually classified into seven basic expressions—fear, disgust, joy, anger, sadness, surprise and contempt—but people’s emotional states are complex and can be further divided into a series of combined emotions, including complex expressions, abnormal expressions and microexpressions; (**b**) Overview of emotionally relevant features of eye movement (AOI 1: The first area of interest, AOI 2: The second area of interest). By collecting data regarding pupil diameter, gaze fixation and saccade, which are three basic eye movement characteristics, their characteristics can be analyzed and counted, including frequency events and special values of frequency event information (e.g., observing fixed frequency and collecting fixed dispersion total/maximum values).

**Figure 3 brainsci-10-00687-f003:**
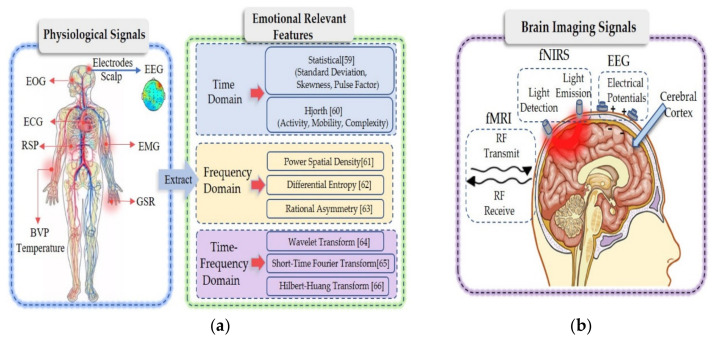
(**a**) Relevant emotion features extracted from different physiological signals. Physiological features and where they were collected are listed, such as EEG, EOG, RSP, EMG, GSR and BVP temperature. We can extract emotionally relevant features from these physiological signals that change over time in three dimensions: the time, frequency and time-43frequency domains; (**b**) Measurement of brain activity by EEG, fNIRS and fMRI. Briefly, we introduced how these three kinds of brain imaging signals collected from the cerebral cortex work with fMRI based on radio frequency (RF) transmit and RF receive; fNIRS is based on light emission and light detection and EEG is based on electrical potentials.

**Figure 4 brainsci-10-00687-f004:**
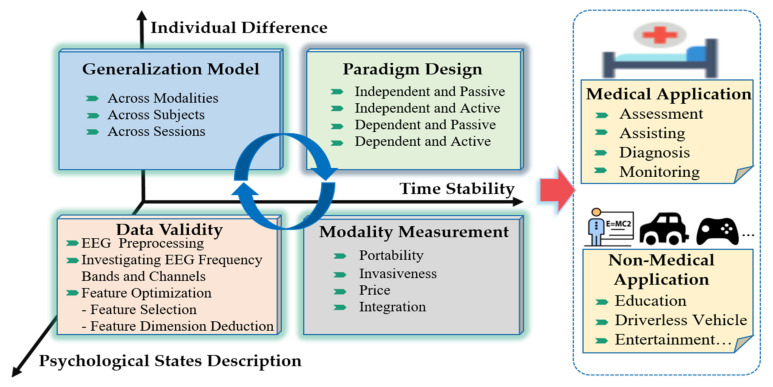
Open challenges and opportunities in multimodal emotion recognition for EEG-based brain–computer interfaces (BCI).

**Table 1 brainsci-10-00687-t001:** Databases based on EEG for multimodal emotion recognition.

Database Access	Modality	Sub.	Stimuli	Emotional Model/States
DEAP [36] www.eecs.qmul.ac.uk/mmv/datasets/deap/	EEG, EOG, EMG, GSR, RSP, BVP, ST, facial videos	32	video	dimensional/valence, arousal, liking
SEED-IV [37] bcmi.sjtu.edu.cn/~seed/seed-iv.html	EEG, eye tracking	15	video	discrete/positive, negative, neutral, fear
ENTERFACE’06 [38] www.enterface.net/results/	EEG, fNIRS, facial videos	5	IAPS	discrete/positive, negative, calm
MANHOB-HCI [39] mahnob-db.eu/hci-tagging/	EEG, GSR, ECG, RSP, ST, Eye-gaze, facial videos	27	video	dimensional/valence, arousal
AMIGOS [40] www.eecs.qmul.ac.uk/mmv/datasets/amigos/index.html	EEG, ECG, GSR	40	video	dimensional/valence, arousal, dominance, liking, familiarity; discrete/basic emotions
EMOEEG [41] www.tsi.telecom-paristech.fr/aao/en/2017/03/03/	EEG, EOG, EMG, ECG, skin conductance temperature	8	IAPS, video	dimensional/valence, arousal
ASCERTAIN [42] mhug.disi.unitn.it/index.php/datasets/ascertain/	EEG, ECG, GSR and visual	58	video	dimensional/valence, arousal
MPED [43]	EEG, ECG, RSP, GSR	23	video	discrete/joy, funny, anger, fear, disgust, disgust and neutrality

Sub.—number of subjects; EOG—electrooculogram; EMG—electromyogram; GSR—galvanic skin response; RSP—respiration; BVP—blood volume pressure; ST—skin temperature; ECG—electrocardiogram; IAPS—International affective picture system; Basic emotions represent the emotions of neutral, happiness, sadness, surprise, fear, anger and disgust.

**Table 2 brainsci-10-00687-t002:** Research on the representativeness of multimodal emotion recognition based on open datasets.

Ref.	Database (Modality)	Feature Extraction	Results (Standard/#)	Innovation
[44]	SEED-IV (EEG + EYE) DEAP (EEG + PPS)	SEED-IV: EEG: PSD, DE in four bands. EYE: pupil diameter, fixation, blink duration, saccade, event statistics. DEAP: preprocessed data.	SEED-IV: 91.01% (avg./4) DEAP: 83.25% (avg. (valence/arousal/dominance/liking)/2)	Using BDAE to fuse EEG features and other features, this study shows that shared representation is a good feature for distinguishing different threads.
[45]	DEAP (EEG + PPS)	EEG + PPS: implicitly extract high-level temporal features.	92.87% (arousal/2) 92.30% (valence/2)	The MMResLSTM learns complex high-level features and the time correlation between two modalities through weight sharing.
[46]	SEED-IV (EEG + EYE) DEAP (EEG + PPS)	SEED-IV: EEG: DE in five bands. EYE: PSD and DE features of pupil diameters. DEAP: EEG: DE in four bands. PPS: statistical feature.	SEED-IV: 93.97% (avg./4) DEAP: 83.23% (arousal/2) 83.82% (valence/2)	The bimodal-LSTM model uses both temporal and frequency–domain information of features.
[21]	DEAP (EEG + PPS)	EEG: power differences, statistical PPS: statistical feature, frequency–domain feature.	83.00% (valence/2) 84.10% (arousal/2)	The parsimonious structure of the MESAE can be properly identified and lead to a higher generalization capability than other state-of-the-art deep-learning models.
[19]	MAHNOB-HCI (EEG + EYE)	EEG: PSD in five bands, spectral power asymmetry. EYE: pupil diameter, gaze distance, eye blinking.	76.40% (valence/3) 68.50% (arousal/3)	Using confidence summation fusion for decision-making fusion of EEG and eye movements can achieve good classification results.
[43]	MPED EEG, GSR, RSP, ECG	EEG: PSD in five bands, HOC, Hjorth, HHS, STFT; GSR: statistical feature; RSP + ECG: energy mean and SSE values.	71.57% (avg./3/positive-negative-neutral)	The novel attention-LSTM (A-LSTM) strengthens the effectiveness of useful sequences to extract more discriminative features.

Ref.—reference; EYE—eye movement; PPS—physiological signals in the DEAP dataset; PSD—power spatial density; DE—differential entropy; BDAE—deep belief autoencoders; LSTM—long short-term memory; MMResLSTM—multimodal residual LSTM; MESAE—multiple-fusion-layer ensemble stacked autoencoder; SSE—subband spectral entropy; HOC—high order crossing; HHS—Hilbert–Huang spectrum; STFT—short-time Fourier transform.

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
