# Peer review of "Advances in Multimodal Emotion Recognition Based on Brain–Computer Interfaces"

_brainsci, 2020, doi:10.3390/brainsci10100687_

Round 1
Reviewer 1 Report
It is clear to me that this paper is substantial and potentially very important. I can see that many people would cite this paper once this is published. But I found this paper extremely painful to read. The main drawback is English. Literally, I have to stop every sentence and paragraph and ponder what exactly the sentence and paragraph mean. I have to say this paper, at least in the current state, is unreadable.
I suspect that few people understand what the authors are trying to communicate and no one can cite this paper. I feel that the authors wrote this paper in their own native language first and someone else who does not know the subject translated the manuscript. That sometimes works but this one is not working.
I recommend the authors to cut unnecessary words, phrases, and sentences as much as possible and streamline the entire manuscript. Avoid inventing new words or phrases. Try to use words you know. I recommend them to use simple and plain language. It would be best if some native speaker proof-reads this manuscript.
Below are some of the problems I encountered (there are lot more but I couldn’t go beyond Sec. 3).
Line 29 “For example, if we can recognize the emotional states of patients with emotional expression disorders, it will be helpful to perform better treatment and nursing measures.
This sentence is hard to understand. Can you simplify?
- Line 46-47 : “In the research of emotion recognition, one of the most important problems is to scientifically describe the emotional state scientifically and construct the emotional description model.”
à “scientifically” is redundant. à “In the research of emotion recognition, one of the most important problems is to scientifically describe the emotional state and construct the emotional description model.”
Line 70 “On one hand, it is difficult to fully reflect emotional state based on single modal …”
? single modal? Single modality? or a single model?
Line 71-72: “On the other hand, some modal information..”
Modal information?? I don’t understand what you mean.
I also don’t understand what you mean by “some modal information is easy to disguise, and it is difficult to get real data.”
Line 174 “Data-level fusion[23], also known as sensor layer fusion. Data-level fusion refers to the direct combination of the most primitive and unprocessed data collected by each sensor to construct a new set of data.”
à Data-level fusion[23], also known as sensor layer fusion, refers to the direct combination of the most primitive and unprocessed data collected by each sensor to construct a new set of data.
Line 191-195 “The open database is convenient for emotional computing researchers to compare algorithms more objectively. Therefore, in the research of emotion recognition, most of the studies focus on the emotion database. In order to enable relevant researchers to more intuitively understand the multimodal emotion data sets based on EEG, we summarize the characteristics of the popular databases in the form of a table (Table 1) and attach the corresponding acquisition links.”
This paragraph is wordy.
How about: The open database is convenient for objective comparison of algorithms; for this reason, much research has focused on the emotion database. To enhance an intuitive understanding of the EEG-based multimodal emotion data sets, we summarize the characteristics of the popular databases in Table 1.”
Line 201-202: “Currently, the research on multimodal emotion recognition of EEG signals is very different in induction materials and acquisition equipment, and the performance of the algorithm is not objective.”
Awkward. how about: “Current research on EEG-based multimodal emotion recognition uses different induction materials and acquisition equipment and lacks objective evaluation of the algorithms.”
Line 203-204: “Therefore, to compare the performance of the algorithm model fairly and objectively, we compare the related research of the open database horizontally “
What do you mean by “horizontally”? You’d better not use the word if you are not sure what you mean.
Sec. 3:
“The core challenge of multimodal emotion recognition is how to better model the internal information of modality and the interaction between modalities. The main purpose of modal internal information mining is to process each modal information independently from other modal information in order to extract the useful information inside the modal. The corresponding information is the interaction between modality, and the expression of emotion is usually accomplished through the interaction between neurophysiology and behavior, so it is very important to capture the relationship between them accurately and make full use of the related information. ”
This paragraph needs streamlining. Very wordy and redundant. Lots of unnecessary information. What do you mean by “emotion is usually accomplished through the interaction between neuropsychology and behavior…”
Emotion is accomplished? You are saying neuropsychology and behavior accomplish emotion. I have no idea what you mean by this.
à“The core challenge of multimodal emotion recognition is to model the internal working of modalities and their interactions. Emotion is usually expressed through the interaction between neurophysiology and behavior; thus, it is crucial to capture the relationship between them accurately and make full use of the related information.
… a lot more.
Reviewer 2 Report
In this work, the authors offer a comprehensive review of brain-computer interface (BCI) for multimodal emotion recognition.
The work is exhaustive, but in my opinion, the review is very discursive and not very focused on technical aspects.
The authors should:
a) integrate more information about preprocessing techinques in emotion recognition;
b) offer more details about the main applications of BCI for emotion recognition.
Minor issues:
- revise english editing;
- please revise expression as "We think that feature optimization can be
divided into two kinds of feature selection and feature dimensionality reduction". Indeed, feature selection and feature reduction are two feature optimization methods. The same for "We believe that not all features carry important information about the emotional state."
